# Imaging of Uveal Melanoma—Current Standard and Methods in Development

**DOI:** 10.3390/cancers14133147

**Published:** 2022-06-27

**Authors:** Małgorzata Solnik, Natalia Paduszyńska, Anna M. Czarnecka, Kamil J. Synoradzki, Yacoub A. Yousef, Tomasz Chorągiewicz, Robert Rejdak, Mario Damiano Toro, Sandrine Zweifel, Katarzyna Dyndor, Michał Fiedorowicz

**Affiliations:** 1Faculty of Medicine, Medical University of Warsaw, 02-091 Warsaw, Poland; m.solnik98@gmail.com (M.S.); natalia_paduszynska@onet.eu (N.P.); 2Department of Soft Tissue/Bone Sarcoma and Melanoma, Maria Sklodowska-Curie National Research Institute of Oncology, 5 Roentgen Str., 02-781 Warsaw, Poland; am.czarnecka@pib-nio.pl; 3Department of Experimental Pharmacology, Mossakowski Medical Research Institute, Polish Academy of Sciences, 5 Pawinskiego Str., 02-106 Warsaw, Poland; 4Small Animal Magnetic Resonance Imaging Laboratory, Mossakowski Medical Research Institute, Polish Academy of Sciences, 5 Pawinskiego Str., 02-106 Warsaw, Poland; mfiedorowicz@imdik.pan.pl; 5Department of Surgery (Ophthalmology), King Hussein Cancer Centre, Amman 11941, Jordan; yyousef@khcc.jo; 6Department of General and Pediatric Ophthalmology, Medical University of Lublin, Chmielna 1, 20-079 Lublin, Poland; tomekchor@wp.pl (T.C.); robert.rejdak@umlub.pl (R.R.); toro.mario@email.it (M.D.T.); 7Eye Clinic, Public Health Department, Federico II University, via Pansini 5, 80131 Naples, Italy; 8Department of Ophthalmology, University of Zurich, 8091 Zurich, Switzerland; sandrine.zweifel@usz.ch; 9Department of Radiography, Medical University of Lublin, 8 Jaczewskiego Str., 20-090 Lublin, Poland; dyndorka@interia.pl

**Keywords:** uveal melanoma, diagnosis, imaging, MRI, PET, CT, SPECT, OCT, ultrasonography

## Abstract

**Simple Summary:**

Uveal melanoma is the most prevalent intraocular tumor in adults, derived from melanocytes; the liver is the most common site of its metastases. Due to troublesome tumor localization, different imaging techniques are utilized in diagnostics, i.e., fundus imaging (FI), ultrasonography (US), optical coherence tomography (OCT), single-photon emission computed tomography (SPECT), positron emission tomography/computed tomography (PET/CT), magnetic resonance imaging (MRI), fundus fluorescein angiography (FFA), indocyanine green angiography (ICGA), or fundus autofluorescence (FAF). Specialists eagerly use these techniques, but sometimes the precision and quality of the obtained images are imperfect, raising diagnostic doubts and prompting the search for new ones. In addition to analyzing the currently utilized methods, this review also introduces experimental techniques that may be adapted to clinical practice in the future. Moreover, we raise the topic and present a perspective for personalized medicine in uveal melanoma treatment.

**Abstract:**

Uveal melanoma is the most common primary intraocular malignancy in adults, characterized by an insidious onset and poor prognosis strongly associated with tumor size and the presence of distant metastases, most commonly in the liver. Contrary to most tumor identification, a biopsy followed by a pathological exam is used only in certain cases. Therefore, an early and noninvasive diagnosis is essential to enhance patients’ chances for early treatment. We reviewed imaging modalities currently used in the diagnostics of uveal melanoma, including fundus imaging, ultrasonography (US), optical coherence tomography (OCT), single-photon emission computed tomography (SPECT), fundus fluorescein angiography (FFA), indocyanine green angiography (ICGA), fundus autofluorescence (FAF), as well as positron emission tomography/computed tomography (PET/CT) or magnetic resonance imaging (MRI). The principle of imaging techniques is briefly explained, along with their role in the diagnostic process and a summary of their advantages and limitations. Further, the experimental data and the advancements in imaging modalities are explained. We describe UM imaging innovations, show their current usage and development, and explain the possibilities of utilizing such modalities to diagnose uveal melanoma in the future.

## 1. Introduction

Uveal melanoma is a neoplasm that originates from melanocytes [1]. In 90% of cases, it develops from the choroid, 7% from the ciliary body, and in 3% from the iris [2,3,4]. In the anterior, the tumor usually localizes in the uveal tract containing the iris. In the posterior, the tumor may localize on the choroid or ciliary body [5].

Tumors are dominated by epithelioid cells, spindle cells, or a combination of them [6,7,8,9]. Depending on the classification, there are seven (subdivided by Callender) or two (subdivided by the Armed Forces Institute of Pathology) types of cells in this tumor [6,10,11]. The simpler classification is used widely. Tumors are, therefore, divided into spindle-cell-type (built with spindle-shaped cells only) and mixed-cell-type (with spindle and epithelioid cells in the tumor). Patients with spindle-cell-type tumors have a better prognosis than those with mixed-cell tumors [12]. UM most often carries mutations in the *GNA11* or *GNAQ* gene [13,14]. Based on the molecular pathology, four subtypes of UM tumors can be distinguished. These molecular abnormalities observed in UM include the number of chromosome 3, arms of chromosome 8q and 6p, two-class expression profiles, and the presence of mutations in the *EIF1AX*, *SF3B1*, or *BAP1* genes [15]. In the American Joint Committee on Cancer classification (AJCC), uveal melanoma is graded according to the tumor basal diameter and thickness, extraocular extension, and ciliary body involvement. Poor prognosis is associate with an advanced AJCC stage [16].

Although the disease is relatively uncommon, it is the most prevalent as a primary intraocular malignancy in adults. Incidence rates of uveal melanoma in various racial or ethnic groups are different. The annual prevalence varies between two per million in southern Europe and eight per million in northern Europe [17]. Differences may be connected with an increased ocular pigment content in southern populations [1]. Patients with uveal melanoma are mostly Caucasian (94–98%), Hispanic (5–2%), and Asian (1% or below), and a very rare incidence (below 1%) has been found in African Americans and American Indians [7,18,19].

The main UM risk factors include a light eye color, fair skin, and the inability to tan [1,20,21]. A fair hair color [1,20] and exposure to ultraviolet light other than welding are not statistically significant risk factors [22]. Other environmental risk factors correlated with UM occurrence may include intense use of sunlamps or sun exposure and chemical exposure (asbestos, pesticides, and formaldehyde), and occupational cooking [21,23]. Established associations have been found between oculodermal melanocytosis, the iris, choroidal nevus, and uveal melanoma. Patients with atypical and common cutaneous nevi and cutaneous freckles are also at risk of UM development [1]. Genetic risk factors include germline mutations in the *MBD4*, *MLH1*, *PALB2*, and *BAP1* genes [9,15,24,25]. UM’s sporadic association has been observed with mutations in the *BRCA1*, *BRCA2*, *FLCN*, *MSH6*, and *CHEK2* genes [15].

UM is most often diagnosed in middle-aged patients (median 60 years old) and is more common in men [19]. The probability of a choroidal nevus transformation into melanoma increases with age. The transformation is an important problem due to the aging of the population in the EU and USA [26,27]. A clinical data analysis of 2355 cases of choroidal nevus revealed a transformation into melanoma of 1.2% at one year, 5.8% at five years, and 13.9% at ten years. In everyday practice, the assessment of a clinical risk factor of UM, such as thickness, the presence of subretinal fluid, symptoms of visual acuity loss, presence of orange pigment, melanoma acoustic density hollow, and tumor diameter, may be of particular importance [27,28].

At the time of diagnosis, clinically evident metastases are detected in 2–4% of all patients [29,30]. Due to hematogenous spread, a liver is the primary and the most common site of metastasis, at 91% [31,32,33]. Other typical sites include lungs (28%), bones (18%), and skin (12%) [31]. It has been indicated that micrometastases are present since the diagnosis because of frequent systemic recurrences and the identification of circulating tumor cells in the bloodstream of patients who do not have clinically detected metastases [34,35]. Even with successful local treatment, over 30% of UM patients develop metastases within ten years [36]. A nonspecific clinical presentation after diagnosing metastatic disease prompts the need for an early and noninvasive identification of UM [37,38,39].

The overall survival of uveal melanoma patients is poor. Once the metastatic stage is diagnosed, the overall survival varies from 3 to 16 months, with a mortality rate of 92% within two years [40]. Tumor size is one of the most important prognostic factors [1,12,16,41]. Small tumors (<3 mm tumor thickness and <10 mm basal diameter), medium tumors (3–8 mm tumor thickness and <15 mm basal diameter), and large tumors (>8 mm tumor thickness and >15 mm basal diameter) have estimated 5-year mortality rates of 16%, 32%, and 53%, respectively. With every one-millimeter increase in thickness, patients have a 5% higher risk of metastatic spread at ten years [12]. Several prediction models have been developed in the last five years based on the genomic expression signatures of primary UM cases. These models allow for predicting overall survival (OS) or metastatic risk in patients with UM [9]. Apart from inherited genetic predisposition, novel prognostic biomarkers have also been established, including proteins involved in the DNA damage response (ATR and ATM), cell death regulation (Beclin-1, BNIP3, HDAC-2, and PARP-1), immune function (BTNL9, PD-1, and PD-L1), genes of the NF-κB pathway (c-Rel, p50, p53, p65, and RelB) receptors (EphA1 and EphA5), cancer stem cell markers (nestin and ABCB5), antioxidant response (PRDX3), kinase signaling (PLK-1), insulin sensitizing (adiponectin), and spermatogenesis (SPANX-C) [9]. Older age at the time of diagnosis [42] and the male gender are also associated with an unfavorable prognosis [16,43]. Due to poor prognosis and limited treatment options in localized and advanced diseases, accurate diagnosis is critical in this group of patients [44,45].

This article aims to analyze the current and future diagnostic options in uveal melanoma, their advantages, and their limitations (Table 1). We focus on the primary disease and complement our analysis with data on the diagnostic technics used for metastatic uveal melanoma.

## 2. Imaging Techniques Currently Used in the Diagnosis of Uveal Melanoma

The standard diagnostic modality for uveal melanoma is its clinical features in an ophthalmological examination supported with an ultrasonography. A biopsy is rarely needed for the diagnosis of uveal melanoma, even though it is very important for prognostication and to obtain material for molecular analyses. Tissue samples obtained with a fine-needle aspiration biopsy (FNAB) are used for genetic analyses, cytologic testing, or FISH, as well as research including a molecular analysis, cell culture, and FFPE archiving. An intraoperative FNAB during plaque surgery may also be conducted [46,47]. Multiple imaging techniques may be employed for UM diagnostics and staging (Table 1 and Figure 1).

The combination of various diagnostic methods has found application in detecting early UM. Techniques such as optical coherence tomography (OCT), ultrasonography, standard wavelength autofluorescence, photography, and visual acuity measurements are used in the multimodal imaging approach. Data analysis obtained with the mentioned methods allows for determining the probability of the transformation of choroidal nevi into melanoma and UM diagnosis. The scheme of proceeding using the above methods is called ‘To Find Small Ocular Melanoma Doing Imaging’ (TFSOM-DIM) [48,49,50].

### 2.1. Ultrasonography

Ultrasonography (US) is a diagnostic tool based on sending and receiving sound waves (with a frequency greater than 20,000 Hz, inaudible to humans) with a probe (containing a piezoelectric transducer), which is both a transmitter and receiver of waves (Figure 2, Figure 3, Figure 4, Figure 5 and Figure 6) [51]. In ophthalmology, two types are used: an A-mode scan to carry out measurements or a movability assessment and a B-mode scan to visualize intraocular structures of different echogenicities as a grayscale view created based on reflected waves [52]. These modes use different frequencies: the A-mode uses 8 MHz and the B-mode uses 10 MHz [53]. 

Uveal melanoma on an A-scan has low to medium reflectivity and presents a positive angle kappa sign, which means that the spikes are high and decrease toward the sclera [1]. On a B-scan, a uveal melanoma is a homogenous mass with a low acoustic profile [54]. Because the eye has a high fluid content, it provides an acoustic window and facilitates a relatively detailed visualization. If the tumor presents a rich vascularization, the pulsations can be registered [55]. The mass can resemble a mushroom shape, synonymous with Bruch’s membrane disruption [56].

An ultrasound can facilitate the assessment of adjacent structure involvement because the extraocular area has a higher reflectivity than the neoplasm, so infiltration is visible as regions of hyporeflectivity beyond the physiological limits of the sclera, which are marked by echoes from the orbital fat [57,58,59]. Thus, US is suitable for assessing tumor extrascleral extension and visualizing lesions even as small as 2 mm [58]. Moreover, it enables the determination of a lesion’s shape and general structure [60]. Due to the properties of ultrasound waves for penetration through tissues with different degrees of pigmentation, it is possible to visualize both highly and nonpigmented tumors [33].

The A-scan-based methodology developed by Ossoinig was utilized in the Collaborative Ocular Melanoma Study (COMS)—the largest study performed in ocular oncology—and has become a standard for the ultrasound diagnosis of choroidal melanoma [61,62,63]. Ultrasonography is commonly used in clinical practice in the diagnosis of uveal melanoma due to availability, no exposure to radiation, and low price (Figure 1). The average cost of US in the world is estimated between USD 155 and USD 760 [64]. An important advantage of US is assessing the lesion’s shape, size, and internal structure [60,65,66]. The mean vascular density can be determined during the study, and that aspect can have a predictive value for the patient [67]. The dimension of the lesion is assessed using a B-scan. However, for a prominence measurement to be correct and avoid overestimation, the transducer must be perpendicular to the lesion, which may be difficult [68]. Considering the limited depth of penetration and the visualization of deep structures, the assessment of the involvement of the extraocular structures with the neoplastic process is difficult [37].

The classic US allows for evaluating the neoplasm of the anterior segment. Still, ultrasound biomicroscopy (UBM), which uses higher frequencies (from 35 MHz to 50 MHz), facilitates the visualization of lesions derived from the ciliary body, tumor progression to posterior regions to the iris, and the differentiation of a cystic or solid nature of the lesion [69] (Figure 3). UBM, with the application of a higher frequency compared to B-scanners, allows for a better axial and lateral resolution of 30 and 60 µm (150 and 450 µm in B-scanner) [70]. Due to the increased resolution in UBM, it is possible to reveal the posterior margin of the tumor and evaluate the disease extension [71]. The disadvantage is the relatively low availability of that method. Several studies revealed the advantage of UBM over OCT in assessing anterior segment tumors, lesions located in the ciliary body, and highly pigmented tumors [72,73].

Scott et al. [74] also compared the effectiveness of US, magnetic resonance imaging (MRI), and computed tomography (CT), and revealed a 100% sensitivity for the ultrasound examination. Martin et al. [75] suggested that the US should be an elementary tool in the initial diagnosis of choroidal melanoma and control during the treatment. Because of the high ultrasound penetration of light through ocular tissues, B-mode imaging is considered a more effective diagnostic process for tumors than anterior-segment optical coherence tomography (AS-OCT) [1]. Retinal detachment, a concomitant condition with uveal melanoma, can be successfully detected with the US because of the accumulation of subretinal fluid which splits the layers (choroid and retina), which are generally seen as integrated, and also enabling the visualization of choroidal growth located behind the damaged retina [76]. Other conditions apart from tumors or retinal detachment as hemorrhage and vascular malformations can be successfully detected with US [77]. This method is also applicable when lesion imaging is disturbed by the media opacity and in patients with hyper mature cataracts that resemble ciliary body melanoma [78].

The diagnostic process of the neoplastic lesion can be complicated when it is located close to the optic nerve, if the ocular muscles are atypically situated, or due to the presence of a vortex vein enlargement [59,75]. The ocular disorders whose sonographic appearance is similar to melanoma are choroidal nevi, metastatic neoplasms, choroidal hemangiomas, disciform lesions, and choroidal hemorrhages [79]. The B-scan method’s tumor thickness is a relevant limitation, because lesions <1 mm could go unnoticed [56]. The size measurements are performed with a high frequency probe of approximately 10 kHz [80]. The significant limitation is that the higher the frequency used, the higher the image resolution, the greater the wave attenuation, and the penetration depth reduces. Due to the accuracy and precision of the method, tumor sizes can be overestimated by the 2D US compared to MR with an average difference value of 1 mm [80] (Figure 4). In research studies, three-dimensional US facilitates the analysis of an extent scale, but it is not performed in routine clinical practice [81].

Color Doppler (CDI) is a noninvasive method that shows the blood flow in vessels with flow direction. The Doppler function (CDUS) could serve as a tool to evaluate the response to radiation treatment because the untreated lesions are characterized by richer vascular patterns than untreated ones [69]. Although ultrasound is widely available, it is an operator-dependent method, and only two-dimensional images can be obtained, other methods allow 3D reconstructions [80]. The disadvantage of this method is its lack of penetration through tissues such as bones, as well as poor imaging of tissues around the perimeter of the wave beam [82].

### 2.2. Optical Coherence Tomography

Optical coherence tomography (OCT) is a diagnostic method analogous to US, based on receiving and processing light waves (within a near-infrared wavelength) to build cross-sectional images (Figure 2) [83]. It uses the interferometry technique; the light is split into a reference beam and sample beam and then reassembled. The echo time delay and reflected or backscattered light intensity are measured to provide images of the tissue microstructure. With depths of 1–2.5 mm into the tissue, OCT can provide images with a spatial resolution of 10 μm or up to 1 μm in ultra-high-resolution OCT (UHR-OCT), the most advanced form of OCT used currently only for academic and research purposes [84]. With a low cost of approximately USD 200, this method is safe, as no adverse events can occur during imaging and no radiation is used [85].

The two main technologies used to create an OCT image are the time domain (TD-OCT) and the Fourier domain (FD-OCT), also known as the frequency domain. The TD-OCT was developed first, and its main limitation came from the need to move the reference mirror with increasing speed to measure the light echoes coming in sequence. the developed later, FD-OCT has a reference arm with a static mirror. It allows for fast scanning as well as improved image quality. Spectral-domain OCT (SD-OCT) evolved from FD-OCT. It significantly increased the acquisition speed up to 100-fold compared to TD-OCT, which in consequence can reduce motion artifacts and enhance the spatial resolution [86].

The OCT is a noninvasive and noncontact method used in the UM diagnostic process, supporting therapeutic decisions and used in treatment control visits (Figure 1) [57]. AS-OCT allows for the real-time imaging of anterior ocular segment elements such as the cornea, anterior chamber angle, sclera, iris, and lens. When it comes to pigmented tumor detection, iris melanoma, iris nevi, and iris melanocytoma can be detected with this method [69]. Conjunctival melanoma is visible on OCT as a hyper-reflective structure covered with a hyper-reflective epithelium regardless of whether the lesion is melanocytic [87]. AS-OCT is a convenient technique used to differentiate conjunctival melanoma from other types of tumors and ocular surface squamous neoplasia [88]. OCT can also efficiently differentiate melanoma and melanocytoma (retinal compound and special grow pattern) [89]. According to come studies, OCT could serve as a technique for the differential diagnosis of conditions such as choroidal metastasis, choroidal osteoma, hemangioma, and retinal tumors [90].

Optical coherence tomography angiography (OCTA) is a new noninvasive imaging technique with the ability to perform a structural and angiographic analysis in vivo, without a contralateral eye dye injection [91,92,93,94]. Recently, Neroev et al. [95] showed that the role of OCTA may be used in the complex diagnosis of early UM and circumscribed choroidal hemangioma for the detection of tumor vessels and the nature of their branching, as well as for a vessel caliber comparison. Additionally, an increase in detecting the tumor’s vessels would allow for the early differential diagnostics of a malignant or benign tumor. It may help in establishing an adequate conserving therapy. In OCT angiography, the choroidal vascular flow rate in choroidal melanoma was significantly lower than in choroidal nevus. Ghassemi et al. [96] found a decreased flow rate of the surface microvasculature (SMV) of choroidal melanoma cases compared with nevi. In their study, Valverde-Megías et al. showed that eyes with choroidal nevus demonstrate a similar central macular thickness (CMT), foveal avascular zone (FAZ) area, and capillary vascular density (CVD) when compared with the contralateral eye. In contrast, eyes with melanoma show an increased CMT, enlarged FAZ, and reduced CVD, particularly related to an increasing tumor thickness [97].

According to some authors [98,99,100], OCTA measurements can provide quantitative biomarkers for the early detection of radiation retinopathy (RR) and/or radiation optic neuropathy following brachytherapy in patients with UM. OCTA provides a quantitative measurement of retinal capillary changes associated with ischemia that correlate with visual acuity and radiation dose, and may predict the future development of radiation-induced retinal toxicity or as a quantitative endpoint to address visual prognosis. Therefore, Matet et al. [101] showed that the visual acuity of eyes with radiation maculopathy is influenced by structural and microvascular factors identified with OCTA, including the FAZ area and DCP integrity.

Jing Yan Yang et al. [102] investigated macular microvascular characteristics imaged with optical coherence tomography angiography (OCTA) in uveal melanoma (UM). Patients were treated with conbercept injections preceded by plaque radiotherapy. According to the results shown by the authors, OCTA may provide a quantitative evaluation of early retinal microvascular changes following radiotherapy. Despite recent developments, image artifacts on OCTA images are commonly encountered and appear to be more frequent in eyes with pathology and poor visual acuity, limiting its use. The recognition of these artifacts might help improve image interpretation and decision making in UM diagnostics [91].

The important limitation of this method is image shadowing occurring in the case of lesions that contain pigment [71]. The light penetration is limited by the sclera or iris epithelium, and AS-OCT is preferable for lesions located superficially and without pigmentation. In other cases, ultrasound biomicroscopy can be a more beneficial choice [72]. However, in a study by Torres et al. [90], all the lesions thicker than 1 mm, which were not detected by US, were revealed using the enhanced depth imaging (EDI-OCT) technique. Pavlin et al. [73] described AS-OCT as a valuable tool for detecting small hypopigmented iris tumors. However, with pigmented iris tumors, OCT visualized only the surface of lesions; their posterior margin was not detected, making a thickness measurement impossible. The same happened with large hypopigmented iris tumors as their image faded out with deeper layers of the lesion. Iridociliary tumors could not be thoroughly examined with AS-OCT as with the provided image. One could only assume the presence of a tumor, but no details can be described adequately.

SD-OCT of the posterior eye segment is a complementary technique for assessing suspicious melanocytic tumors in the choroid. Despite the difficulty in imaging deeper tissues within choroidal melanoma, the anterior tumor surface and secondary retinal changes can be documented with this method. Materin et al. [103] summarized how findings of subretinal fluid could help distinguish between choroidal nevus and melanoma, since the presence of subretinal fluid is one of the high-risk factors for the growth of a tumor. On the other hand, Vishnevskia-Dai et al. [104] investigated the different characteristics of the anterior choroidal surface between choroidal melanoma and metastases. Overall, choroidal melanoma presented a regular and smooth anterior surface on OCT, while choroidal metastases demonstrated lobulated and irregular surfaces.

With the introduction of the enhanced depth imaging OCT (EDI-OCT), normal choroidal anatomy and choroidal diseases can be visualized in detailed and measurable images [105]. The current axial resolution of EDI-OCT is approximately 3 to 4 μm (for comparison, the axial resolution of TD-OCT is 10 μm, and US is 50–100 μm) [105]. In a study by Torres et al. [35], all the lesions thicker than 1 mm, which were not detected by US, were revealed using the EDI-OCT technique. Measurements of a tumor’s maximum diameter and thickness were possible only in lesions <9 mm in diameter and <1 mm in thickness. The study also suggests using EDI-OCT as an additional tool in the differential diagnosis of ocular tumors. EDI-OCT’s specific characteristics, including a highly reflective band within the choriocapillaris found in melanocytic tumors or the enlargement of the suprachoroidal space found in choroidal metastases, can help in tumor identification. Overall, EDI-OCT allows for a finer resolution of the choroidal findings with associated retinal features (subretinal fluid, subretinal lipofuscin deposition, and shaggy photoreceptors).

Shields et al. [106] evaluated 37 eyes with small choroidal melanomas (thickness of 3 mm or less) using EDI-OCT in a retrospective comparative analysis. The authors noted that compared with EDI-OCT, the tumor thickness in US was overestimated by 55% (2300 μm with US vs. 1015 μm with EDI-OCT). The discrepancy was most likely caused by the moderately gross estimation using US calipers and by the inadvertent inclusion of an overlying retina with the choroidal tumors’ anterior surface. The features of small choroidal melanoma presented on EDI-OCT included optical reflectivity along the anterior surface with partial (73%) and complete (27%) optical shadowing more deeply within the tumor. Moreover, compared with a choroidal nevus, an increased number of abnormal photoreceptors, intraretinal edemas, and irregularities of the inner plexiform layer were found in small choroidal melanoma.

However, EDI-OCT is not recommended for all patients. Shah et al. [107] concluded that the EDI-OCT technique provided suboptimal images in tumors with a larger basal diameter (>5 mm), in older patients (>60 years), as well as in the extramacular location of the tumor. Those findings support the use of this modality in small choroidal melanomas and the use of US in larger tumors.

### 2.3. Fundus Imaging (FI)

Indirect ophthalmoscopy is the primary and most important method to diagnose UM (Figure 2) [108,109]. In this technique, a direct ophthalmoscopy image of the patient’s retina appears directly on the observer’s retina [110]. This observation allows for determining several defining features, such as the size of the tumor, retinal detachment (in tumors greater than 4 mm in thickness), the presence of pigmentation, its location (distance to the foveola and optic disc), extrascleral extension towards the anterior direction, and the involvement of ciliary body [15,108].

Fundus photography (FP) performed initially allows for assessing retinal pathology. With FP, an ophthalmologist can examine the central retina, macula, optic nerve, and, after mydriatics, the peripheral retinal area. It is a primary method of retinal imaging [111]. Choroidal melanoma looks like a dome, mushroom-shaped, or diffuse form. The color of the tumor is gray to greenish-brown. Retinal detachment, necrosis, or the atrophy of retinal pigment epithelium cells may occur [112]. Fundus photography allows the detection of small lesions with a basal diameter lower than 3 mm [113].

Nowadays, wide-angle systems can capture up to 200° by utilizing montage images. Wide-angle systems could be used to observe shallow margins of tumors and are precise as USG, although these systems are characterized by low sensitivity in detecting lesions localized anteriorly to the equator [112,114,115,116].

Imaging of fundus with photography has evolved from analog film-based to digital pictures. Nowadays, images are obtained with ocular fundus cameras with digital photography. Recently, smartphone-based fundus imaging has been developed (a smartphone connected to a digital fundus camera). Compared to conventional digital fundus photography, smartphone-based fundus imaging is a low-cost alternative to imaging the retina. This equipment requires less training time for the person performing the diagnosis. Obtained images possess a good quality comparable to computers [117,118]. Due to portability, better availability, noninfrastructure requirements (continuous power supply), and obtainment by nonspecialists personnel (images can be sent for further analysis to retina specialists), this technique may improve the study of more patients in the future, especially in communities with a less developed medical infrastructure in primary care [119].

Photographs of the retina enable the detection of small lesions, including uveal melanoma. To diagnose UM tumors more precisely, additional techniques such as OCT or US are additionally used [56]. Alone fundus photography may be useful to monitor tumor growth, assess therapy treatment, or arise recurrences [57,69,112].

### 2.4. Fundus Fluorescein Angiography, Indocyanine Green Angiography, and Fundus Autofluorescence

Fundus fluorescein angiography (FFA) is an imaging technique used to examine circulation in the retina and choroid, and provide information about the retinal pigment’s blood–retina barrier integrity and characteristics of the epithelium (Figure 6). The intravenous administration of fluorescein dye (sodium fluorescein) is followed by illuminating the retina with blue light at 465 nm. A photon of light is absorbed by an electron at its resting stage, causing its excitation. Further, as the electron relaxes to the resting stage, a photon of yellow–green light is released. This process is captured by a series of quickly obtained photographs or a film taken after dye injection [120,121].

In large uveal melanomas, intrinsic tumor circulation is paired with choroidal circulation. This double circulation, later leakage, and hot spots resulting from pinpoint leakages from the retinal pigment epithelium (RPE) are characteristic features of FFA in uveal melanomas. However, this imaging modality alone lacks diagnostic accuracy for UM detection [69,108]. Meyer et al. [122] assessed the diagnostic accuracy of FFA in the diagnosis of choroidal melanoma and obtained results between 17% and 75%, with 58% on average. Nevertheless, bundling multiple images acquired with FFA, OCT, and fundus autofluorescence could help increase the accuracy of diagnosis. On the other hand, FFA is a valuable method for differential diagnosis, as it can detect a hemangioma, hemorrhage, or choroidal detachment [69,123]. FFA is also used in follow-up after brachytherapy to detect complications arising from radiotherapy, such as radiation retinopathy and maculopathy [57,109].

Indocyanine green angiography (ICGA) is similar to FFA, except that the fluorescein dye used in this method is indocyanine green. Its peak absorption is at approximately 790 nm, with a peak emission of 835 nm. These properties allow the penetration of retinal layers, melanin, and macular pigment, making it possible to visualize structures beneath RPE. Indocyanine green binds to plasma protein in approximately 98% of cases, making it impossible for the dye to exit the bloodstream through choriocapillaris fenestrations, allowing the visualization of not only large and medium vessels, but choriocapillaris as well [124,125].

ICGA provides a greater visualization of tumor vasculature, as it can imagine the microcirculation of uveal melanoma, especially when there is overlying blood [15,108]. Variations of ICGA (hypofluorescent, isofluorescent, and hyperfluorescent) depend on tumor thickness and the degree of pigmentation [124]. These nonspecific patterns disqualify ICGA as an accurate diagnostic modality; however, ICGA can potentially differentiate pigmented and nonpigmented choroidal melanomas and choroidal hemangiomas. Nonpigmented tumors compared with pigmented tumors show an earlier onset of fluorescence (<1 min vs. 3 min), and choroidal hemangiomas reveal characteristic findings on ICGA, including an early onset of fluorescence and early maximum fluorescence, followed by a “washout” of the dye in later frames [69,126]. The application of lens coated with an antireflection substance and an indium iodide lamp allows the obtainment of a 7.4 µm resolution [127]. It should also be mentioned that ICGA can induce phototoxicity on RPE cells, and unnecessary use should be avoided [128].

Fundus autofluorescence (FAF) is an in vivo imaging technique that allows for the visualization of the stimulated emission of light coming from fluorescence chemical compounds (fluorophores) found in the ocular fundus with a fundus camera containing specific filters [123]. The most common fluorophore is lipofuscin, which accumulates in RPE cells and is seen as an orange pigment on indirect ophthalmoscopy [129]. Secondary epitheliopathy is common in malignant choroidal tumors, causing the shedding of photoreceptor outer segments. Combined with high phagocytic activity, this process increases lipofuscin accumulation seen in uveal melanomas [123,129].

One of the predictive risk factors of small melanomas is an overlying orange pigment visualized with FAF. Therefore, FAF could play a role in the early detection of small choroidal melanomas and differentiation between choroidal melanoma and nevus [130]. Choroidal melanomas demonstrate clumps of hyperautofluorescence, whether most of the choroidal nevus is isoautofluorescent or hypoautofluorescent [15,108,129]. Additionally, blue-light FAF can be used to confirm the proper location of radioactive plaque used in local treatment [15]. Finally, all of these fluorescence-based modalities are dependent on the patient’s ability to stay still to avoid motion artifacts and experience the interpretation of acquired images [120,123]. Fundus autofluorescence images can be obtained through scanning with a laser ophthalmoscope with blue-light excitation or a fundus camera with appropriate filters. These methods allow a 5 or 14 µm per pixel resolution, respectively [131].

### 2.5. Magnetic Resonance Imaging

Images obtained with MR depend on the imaging modality, e.g., PD-weighted (PDw, proton-density-weighted), T1-weighted (T1w, longitudinal-relaxation-time-weighted), T2-weighted (T2w, transverse-relaxation-time-weighted), or diffusion-weighted imaging [132,133]. This method is also highly suitable in experimental settings (Figure 7), e.g., for studying animal models of uveal melanoma, allowing for the qualitative evaluation of the images and quantitative measurements of the ocular dimensions [134]. The value of the static magnetic field generated by the devices in clinical practice is typically 1.5 or 3 Tesla, but in experimental settings, the values may exceed 7 Tesla [37]. The greater the magnetic field used, the more profitable the signal-to-noise ratio and the shorter the acquisition time [132,135]. MRI was shown to provide a more accurate measurement of tumor dimensions than US; in some cases, MR-based diagnosis could allow for eye-preserving therapy [68].

A good soft tissue contrast creates a favorable condition to visualize intra- and extraorbital compounds, tumor localization to assess tumor dimension, and the involvement of adjacent tissues [136,137]. Nowadays, acquired images influence therapeutic decisions and the evaluation of tumor advancement, prognosis, and response to treatment [138]. The irrefutable advantage of MR is that the patient is not exposed to ionizing radiation and avoids its potential complications. GrechFonk et al. estimated the average cost of MRI at EUR 200–EUR 1000 (the price is lower in Asia, Australia, and Europe, and higher in the USA; Table 1, Figure 1). They suggested that, although the price is not lower than ultrasonography, it is still significantly lower than the possible costs of therapy [80].

The presence of orbital adipose tissue enables a relatively good tissue contrast [132]. The characteristic features of uveal melanoma obtained in MR (Figure 8) compared to the vitreous (because of the high-water content in the vitreous body, it is hypointense in T1w and hyperintense in T2w) include hyperintensity on T1-weighted images, hypointensity on T2-weighted images, hyperintensity on PD images, and homogenous enhancement with a gadolinium contrast [138,139,140]. T1- and T2– relaxation times are shortened by unpaired electrons of melanin, and this property correlates with the quantity of melanin [132,141,142,143]. An uneven melanin distribution can contribute to an inhomogeneous intensity character, similar to disturbing the vascular system of a tumor [132,137]. However, an image analysis can be troublesome in the case of amelanotic melanoma, because the T2 relaxation time cannot be shortened, so this technique cannot distinguish the amelanotic melanoma from melanotic melanoma [59,144].

In certain situations, the recognition of uveal melanoma may be problematic due to other medical conditions that look similar. For example, choroidal metastases could mimic a uveal melanoma [138]. Furthermore, the differentiation between uveal melanoma and other pathologies such as vitreous hemorrhage is also challenging based on T1w or T2w imaging only [138]. Fresh blood can be isointense or hypointense on the T1w or T2w images, and the hyperintensity can occur after a few weeks [144]. The differences in blood appearance in MRI depend on the oxygenation of hemoglobin, oxy-, deoxy-, or methemoglobin present at different stages of hemorrhage. Because of the neoplasm extension, a common concomitant condition is a retinal detachment, which can be confirmed with a no-contrast enhancement reaction (unlike the tumor). Still, without using a contrast agent, these lesions look very similar [138].

In a study by Mafee et al. [144], among 21 patients with suspected uveal melanoma and other complications such as hemangioma (less contrast enhancement than melanoma), choroidal, or retinal detachment, magnetic resonance imaging was an effective technique. Stroszczynski et al. [145] suggested that MR could be a more favorable method than computed tomography. However, compared to CT, MRI is less appropriate for the visualization of calcifications [146]. According to Rescan et al., MRI is characterized with 100% sensitivity in detecting scleral involvement and extrascleral propagation with 50% and 89% specificity, respectively [147,148]. A 3T magnetic resonance allows for the acquirement of a resolution of approx. 800 µm, and a 7T apparatus allows for a 500 to 650 µm resolution [80,149]. More accurately, 32 μm was obtained with a high-spatial-resolution MRI at 9.4 T [150].

Artifacts resulting from involuntary eye movements are a significant obstacle that deteriorate image quality [151]. Drifts, tremors, saccades, or blinking may occur; thus, it is important to restrict movement. One of the possible countermeasures is the application of retro- or parabulbar anesthesia [152]. Due to the permanent opening of the eyelids during the imaging process, different devices that aim to ensure the immobilization of the globe contribute to magnetic field disturbances in the path between the air and eye surface [153]. The best solution to limit artifacts would be to shorten the acquisition time, but it would negatively affect the image quality [138]. The surface coil is placed on the patient’s head to improve the resolution properties but with a relatively low accuracy in visualizing the deeper orbital tissues, because the signal intensity decreases with the distance from the coil [138].

Outside of the assessment of primary tumors, MR imaging is used to detect distant metastases, especially in the liver. Marshall et al. [154] showed that a six-monthly MRI could detect approximately 90% of hepatic lesions before they became symptomatic, enabling early treatment in the group of patients with high-risk uveal melanoma. Similarly, Servois et al. [155] assessed MRI and ^18^F-FDG PET applicability in the preoperative staging of hepatic metastases from uveal melanomas. The study showed the superiority of MRI over ^18^F-FDG PET in this regard, with differences marked especially for lesions smaller than 10 mm. Correspondingly, in the study conducted by Francis et al. [156], MRI had high accuracy in the detection of metastatic liver lesions in newly diagnosed uveal melanoma patients.

Additionally, an eye MRI serves as an important tool in the choice of the therapeutic method and provides information about anatomical relationships before brachytherapy, proton beam, and stereotactic radiotherapy [56,157]. Nevertheless, when considering patients with less than 6 mm or more than 8.5 mm tumor thickness, it turned out that MRI does not enable the introduction of a significant modification into therapy selection [80].

MR imaging can also be essential in patients who have undergone a vitrectomy. In these cases, replacing the vitreous body with a silicon oil (SiOil) tamponade causes difficulties for UM follow-up, because in imaging methods such as US, sound waves are reflected at the SiOil–water interface [158]. By principle, SiOil does not hinder MR imaging [147,148], although some powerful artifacts may appear in the conventional protocols due to the off-resonance of the SiOil. Dedicated MRI protocols with 3 T and 7 T MRI might be able to provide high-resolution images of vitrectomized eyes with a SiOil tamponade [158].

The basic protocol needed to assess the tumor dimensions includes T1- and T2-weighted images before and after the application of a gadolinium contrast agent, and lasts approximately 20 min [159]. The other sequences could be used to gain additional and more precise information. The GE (gradient echo) sequences characterize the neoplasm structure, and the SE (spin echo) and fat suppression sequences image the scleral boundary [138]. The multiplanar reconstructions provide measurement possibilities in various planes and reveal possible propagation beyond the sclera similar to postcontrast T1-weighted scans with fat suppression. The isotropic sequences visualize the tumor boundaries [138]. The reconstructions of 3D TSE (turbo spin echo) sequences are desirable in disturbances due to eye motion. Another modality is diffusion-weighted imaging (DWI), which is based on measuring water molecules’ Brownian motions depending on the cellularity or presence of edema to generate a contrast on MR images. DWI may differentiate benign or malignant tumors and use the ADC (apparent diffusion coefficient) to analyze a prognostic response to treatment with the proton beam [157,160]. A significant drawback of DWI is its high susceptibility to motion artifacts. Therefore, future studies are desirable to develop a solution capable of reducing artifacts and shortening the acquisition time.

Other promising MR modalities in UM diagnostics include perfusion-weighted imaging with gadolinium CA, which is clinically used in almost all oncological MR exams and has a specific UM value. Such an approach can differentiate between UM and retinal detachment [157,161]. Three-dimensional imaging may also be superior to US [68] and for radiotherapy planning [162]. Another remarkable approach is diffusion-weighted imaging used for differential diagnosis [160] and treatment follow-up [163].

### 2.6. Computed Tomography

CT uses X-ray beams that cross the human body and receive the detectors. Detectors receive the signal and convert it into electrical pulses sent and analyzed by a computer. As a result, data obtained with the scanning process contain cross-sectional images of the human body [164,165]. Modern CT systems develop several special techniques. MDCT methods (image segmentation using manual or semiautomatic approaches) allow to determine the volume of objects on CT scans. It may apply in diagnosing osteoporosis, the loss of skeletal muscle, and measuring adipose tissue or organ volumes. CT allows for the localization of a lesion and its impact on adjacent tissues. Knowing the precise location facilitates planning a surgical approach. CT angiography (CTA) allows for assessing a vascular tree applicable in stroke evaluation, aneurysm location, size, or neurosurgery. MDCT combined with PET can highlight amyloid depositions in imaging Alzheimer’s disease [164,166,167].

CT is rarely used in UM diagnosis. Other techniques such as MRI or US possess a better precision and are commonly used [15]. When using CT scans, it is difficult to distinguish UM from retinal detachment or exudative macular degeneration. CT images give unambiguous results when examining epi- and extrascleral growth (e.g., in one case not lesions, but macular degeneration-type Junius–Kuhnt revealed with funduscopy). CT allows to easily distinguish vitreous bodies from the lesion (due to possessing a high radiographic density), but a small uveal melanoma lesion was impossible to detect. It is difficult to precisely determine the size of a tumor and the minimum measurement of a lesion. Measurements of tumors were very inaccurate (too small, too big, or the tumor was not visible) compared to the US or histological examinations [168]. In CT scanning systems for the eye and orbit, an approximately 1 mm thick slice scan (in most CT scanners, the spatial resolution is 500–625 µm despite being thinner than a 2 mm tumors) could not be observed [167,169,170].

CT is also nonspecific. Lesions are difficult to recognize due to the high attenuation of the signal by the choroid. MRI imaging is better than CT because melanin has intrinsic T1 and T2 shortening effects, distinguishing the presence of tumor lesions [143]. CT has been used to confirm diagnostics performed previously with echography or US to confirm extrascleral and extraocular extension [74,75,171]. It also may be useful to observe bony orbital expansion or scalloping. It may also show small calcifications [172]. CT may help in the diagnostic process in eyes with a secondary vitreous hemorrhage or cataract [56].

Computed tomography may be used for accurate staging when metastasis occurs. CT and PET are used to assess liver and extra-liver (e.g., bone and lymph nodes) metastatic choroidal melanoma [173].

### 2.7. Single-Photon Emission Computed Tomography

Single-photon emission computed tomography (SPECT) uses gamma-emitting radiopharmaceuticals delivered intravenously into patients’ bodies and records their emission from many angles with a nuclear camera to provide 3D images [174]. Radionuclides used in SPECT emit one gamma-ray photon, differentiating them from radionuclides used in positron emission tomography (PET), which emit two photons created after a positron–electron encounter [175]. The nuclear cameras rotate around the patient, while gamma photons radiating from the patient’s body pass through the collimator and finally hit the detector plane. After the detectors collect all the data (the location of the interaction and photon energy), they can create an image of the tissues [176]. The spatial resolution depends mainly on the type of collimator used, as it can improve with a collimator with smaller and longer holes. A 9.3 mm (FWHM) resolution can be achieved using all/general-purpose collimator types [174]. SPECT using Technetium-99m-MIBI accumulation in tumors to evaluate ocular malignant lesions in patients allows the detection of lesions in sizes 9.5 to 12 mm [177].

Similar to PET, gamma photons are emitted from high radionuclide accumulation sites. Radiotracers used in SPECT include ^99m^Tc, ^201^Tl, ^67^Ga, or ^123^I. They all have a long half-life time that varies between several hours and days, allowing a long imaging time and making the diagnostic accuracy highly time-dependent [176].

N-isopropyl-p-[^123^I]iodoamphetamine (^123^I-IMP) is a radiopharmaceutical used initially as a tool for blood perfusion imaging. Later, it was found that ^123^I-IMP accumulates in areas with increased melanin production [178]. Goto et al. [179] found that ^123^I-IMP SPECT is a suitable diagnostic method for detecting uveal melanoma with atypical manifestations or for its diagnosis when ocular complications (cataract, hemorrhage, etc.) are present. A significantly high intake of ^123^I-IMP in an adequate tumor location was observed in 25 out of 27 (92.6%) patients with suspected uveal melanoma who did not undergo any treatment. The smallest found lesion was 3 mm × 4 mm. In the same study, 35 out of 36 (97.2%) patients without significant ^123^I-IMP accumulation were diagnosed with different intraocular disorders. In another study,^123^I-IMP SPECT was performed 24 h after the intravenous administration of radionuclide in a group of 19 patients. In all the 12 (100%) patients with an increased uptake of ^123^I-IMP, uveal melanoma was diagnosed. However, only two of them were SPECT-positive three hours after the ^123^I-IMP application. In the remaining seven results, one was falsely negative (85.7%). The ^123^I-IMP SPECT method was also shown to be more sensitive and accurate than the 18-fluoro-2-deoxy-D-glucose (^18^F-FDG) PET in uveal melanoma diagnosis [180].

Abe et al. [178] evaluated the degree of usability of ^123^I-IMP SPECT using the uptake index (UI) and increased rations (IRs) as the defining tools. The accuracy of the ^123^I-IMP SPECT method improved with time after radionuclide administration. At the 48 h time point and with a UI cut-off value of 1.90, the sensitivity and accuracy were 100% and 95%, respectively. Moreover, calculated UI and IRs in uveal melanoma patients were significantly higher than in the group of patients without uveal melanoma. In a recent study, Yamazaki et al. [181] assessed SUV in SPECT/CT hybrid images to provide comparable UI evaluation results. Moreover, the SUV-based method evaluated at a 6 h time point can forecast the degree of ^123^I-IMP uptake at a 24 h time point, and is more predictable than the UI evaluation method.

SPECT is superior to PET in costs, as SPECT has been valued at USD 1900 [55] on average, and its availability [182]. However, due to the use of radionuclides in SPECT and even additional radiation coming from X-rays in SPECT/CT, an increased cancer risk is a significant hazard [183].

### 2.8. Positron Emission Tomography/Computed Tomography

Positron emission tomography is a diagnostic imaging technique that visualizes areas based on their affinity and biodistribution of administered radiotracers [184]. After administering a radiopharmaceutical, IT is distributed in the tissue, emits a positron that interacts with an electron, and creates two photons emitted in virtually opposite directions [185]. It allows for the visualization as well as the quantification of the stated biodistribution in a three-dimensional reconstruction [184]. PET cameras can provide images with a spatial resolution of approximately 2.4 mm full-width at half maximum (FWHM) [186]. The most commonly used radiotracer, ^18^F-FDG, allows for the detection of tumor cells, because of their high glucose uptake [187].

Consequently, oncology has widely used PET to detect tumors and metastases, evaluate treatment response, and detect recurrences due to its current availability and effectiveness [29]. However, high glucose avidity is also related to inflammation, infection, or trauma. Some treatment options, such as radiotherapy, can also cause a high glucose uptake up to 3 months after the procedure [188]. With the combination of CT and PET, information about metabolic abnormalities is anatomically accurate, making this method much more accurate [37,187].

For a semiquantitative image analysis, the standardized uptake volume (SUV) is used. It shows the correlation between tracer activity in tissue (microcuries per gram) and injected radiotracer dose (millicuries) as well as patient weight (kilograms). Typically, an SUV value above 2.5–3.0 indicates malignant tumors. The SUV varies based on the time between the radiotracer administration and PET imaging, making it important to standardize the time interval in PET/CT scans, especially when used for a treatment response evaluation [189].

Reddy et al. [190] reported that the AJCC T2-stage melanomas had a 33% primary tumor detection rate, 75% for the T3-stage, and none of the T1 tumors (with the SUV cut-off value of 2.5). SUV correlated positively with the tumor volume in UM [191]. Matsuo et al. [192] observed a high glucose uptake in nodular choroidal melanomas in a different study, but diffusely infiltrating tumors were not detected. It was also shown that a higher SUV_max_ correlates with a greater thickness of the UM tumor [183,192]. McCannel et al. [193] found correspondence between metabolic activity on PET/CT scan and Monosomy 3 (54% sensitivity and 100% specificity). The correlation between positive PET/CT imaging and increasing the tumor size along with Chromosome 3 loss was also described by Papastefanou et al. [194]. A high SUV value was found in older patients with lesions presenting larger diameters linked to poor prognosis and a greater metastatic risk [191]. According to those findings, PET/CT seems to be a useful tool for predictive purposes, but not for primary tumor diagnosis, as it may not be able to detect small uveal melanomas or differentiate them from uveal nevi [184,190].

The main role of ^18^F-FDG in PET/CT is known to be the detection of UM distant metastases [178]. Both Klingenstein et al. [195] and Kurli et al. [173] reported 100% sensitivity and 100% specificity of this method in detecting liver metastases in UM patients. However, in a different study, Strobel et al. [196] reported that PET/CT identified only 16 of 27 (59%) known liver metastases. Freton et al. [29] described a whole-body 18F-FDG PET/CT as a valuable method for initial staging in patients with UM in a retrospective study. Hepatic metastases observed on PET/CT scans were confirmed via biopsies, suggesting a 100% positive predictive value. Extrahepatic lesions (lungs, bone, lymph node, brain, and spleen) were also found even though blood tests and other imaging modalities did not indicate the presence of metastatic disease in those sites. Furthermore, PET/CT seems to be valuable in assessing an early therapy response. In response to the administration of chemotherapy, PET/CT showed decreased metabolic activity in lesions, even though their size remained the same according to MRI [197]. However, DW MRI can also show an early treatment response [163].

PET/CT is an expensive imaging technique as its cost in the US varies between USD 429 and USD 2933 (average of USD 2227; Table 1, Figure 1) [64]. As well as other imaging methods using radiotracers, PET/CT scanning is associated with a significant radiation dose and can increase secondary cancer risk concerns in patients, so the risk–benefit ratio should be explained to UM-diagnosed individuals [198].

## 3. New and Experimental Imaging Techniques with Potential in UM Diagnostics

### 3.1. New Radiolabeled Radiotracers in PET

As of now, ^18^F-FDG is the most common radiotracer used in PET in oncology imaging. However, false-negative results are seen in well-differentiated tumors or lesions with low metabolic rates, and false-positive results can be observed in areas with active inflammation. Over the years, the understanding of cancer cell biology has become more advanced, and processes outside the glucose uptake have been discovered. Therefore, new non-FDG radiolabeled radiotracers have emerged to overcome ^18^F-FDG limitations and better imaging capabilities for certain types of cancer [199,200].

The oldest non-FDG tracer is bone-specific sodium fluoride labeled with fluorine 18 (^18^F-NaF). Fluorine ions bind to the surface of hydroxyapatite proportionally to the bone remodeling and bone blood flow, indicating undergoing osteolytic or osteoblastic processes [199,201].

All of this led to this radiotracer being used in PET alone or PET/CT combination to detect and assess skeletal metastases (Table 2). Most studies were conducted on ^18^F-NaF-PET concerning detecting bone metastases in prostate, breast, lung, and thyroid cancer [193,201]. In a meta-analysis, Liu et al. [202] compared ^18^F-NaF PET/CT and ^99m^Tc-methylene diphosphonate (MDP) planar bone scintigraphy (BS) in bone metastases diagnosis. Based on the patient, sensitivity was higher when equivocal results were positive (96% vs. 93%), but specificity was higher when equivocal results were reported as negative (95% vs. 93%). Based on the lesion, both sensitivity and specificity showed similar results to those based on the patient basis. Compared to ^99m^Tc-MDP BS, ^18^F-NaF PET/CT showed a higher diagnostic accuracy in bone metastases detection with superior sensitivity and specificity. Similar results were shown in a meta-analysis by Shen et al. [203].

Although this modality has not yet been used directly in the imaging of uveal melanomas, it can be applied for bone metastases detection, which is the most common site outside of liver metastases, or for assessing the extrascleral extension of lesions in the near future.

In uveal melanoma, available imaging methods can hardly detect stage I and II melanomas and small distant metastases. Consequently, new specific radiotracers that can bind to melanin have been developed, including radiolabeled antibodies, benzamide (BZA), and benzamide analogs [204,205]. Although the studies described below focused on cutaneous melanoma, highly specific melanin tracers should also be considered for uveal melanoma diagnosis and UM distant metastases.

One of the more recently introduced melanin-targeted radiotracers is 5-bromo-N- -(2-[diethylamino]ethyl) picolinamide labeled with ^18^F (^18^F-5-FPN). A high affinity characterizes this benzamide derivative to melanin and quickens its renal clearance [204,205]. In the preclinical study conducted by Feng et al. [204], the potential of ^18^F-5-FPN PET in the diagnosis and staging of melanoma was evaluated in vivo in B16F10 tumor-bearing mice. ^18^F-5-FPN PET displayed high effectiveness in detecting primary tumors and simulated small metastases (1–2 mm) in the lungs.

Compared to ^18^F-FDG PET, the tumor-to-background ratio in ^18^F-5-FPN PET was ten times higher. Nevertheless, a significant disadvantage of ^18^F-5-FPN PET is the hepatic retention, which can reduce the efficacy of this method in detecting hepatic metastases. Similar results were obtained in the study by Wang et al. [205]. Models of lymph node and pulmonary metastases were created in C57BL/6 mice and then evaluated with both ^18^F-5-FPN PET and ^18^F-FDG PET. On ^18^F-5-FPN PET scans, small metastases in both lymph nodes and lungs were more visible, and the tumor-to-background ratio was significantly higher than in ^18^F-FDG PET scans.

The evaluation of the ^18^F-6-fluoro-N-[2-(diethylamino)ethyl] pyridine-3-carboxamide (^18^F-MEL050), melanin-specific benzamide analog for the imaging of primary and metastatic melanomas was conducted by Denoyer et al. [206]. In that study, an ^18^F-MEL050 PET scan was performed on melanoma model C57BL/6 mice inoculated with B16-F0 and B16-BL6 murine melanotic melanoma cell lines. ^18^F-MEL050 clearly delineated melanoma lesions with rapid background washout and a higher tumor-to-background ratio than ^18^F-FDG. Rizzo-Padoin et al. [207] confirmed these data and reported that ^18^F-MEL050 PET/CT could detect submillimeter pulmonary metastases not found on ^18^F-FDG PET/CT. Denoyer et al. [208] reported that lymph node metastases could be identified with perilesional ^18^F-MEL050 administration.

To overcome the significant hepatic retention of ^18^F-5-FPN, its modification ^18^F-N-(2-diethylaminoethyl)-4-(2-[2-(2-fluoroethoxy)ethoxy]ethoxy)pyridine (^18^F-PEG3-FPN) was recently introduced. ^18^F-PEG3-FPN was proven to have a lower liver uptake than ^18^F-5-FPN, while still demonstrating a high affinity to melanin. Hepatic and pulmonary metastases in the mouse model were detected with ^18^F-PEG3-FPN PET with a high tumor-to-background ratio, suggesting this imaging method for diagnosing primary and metastatic malignant melanoma [209]. The clinical application of ^18^F-PEG3-FPN PET/MR and PET/CT is going to be evaluated in patients with clinically suspected or confirmed melanoma in the recruiting clinical trial with a study completion date estimated at the end of 2023 (NCT04747561). Other radiolabeled melanin-specific tracers assessed in studies include ^18^F-labeled picolinamide (NCT03033485) and a new experimental drug ^18^F-SKI-248380, a dasatinib derivative (NCT01916135). With high diagnostic potential shown in cutaneous melanoma cases, studies of these tracers should be expanded to uveal melanoma patients in forthcoming years.

### 3.2. PET/MRI

Discussions about combining PET and MR imaging started in the 1990s alongside the proposal of PET/CT. A great soft tissue contrast, a low dose of radiation, various imaging sequences, and possible simultaneous image acquisition suggested its promising role in oncological, neurological, and cardiovascular applications [210].

PET/MRI showed superiority over PET/CT in liver metastases detection, especially for detecting lesions smaller than 1 cm [211]. Donati et al. [212] compared liver metastases detection in PET/CT, gadoxetic-acid-enhanced MRI, and retrospectively fused images of PET and MR in 37 patients. The sensitivity was significantly higher for PET/MRI (93%) compared to PET/CT (76%). However, in gadoxetic-acid-enhanced MRI (91%), the sensitivity was not significantly different from PET/MRI. In the study conducted on 70 patients by Beiderwellen et al. [213], PET/MRI showed a higher diagnostic accuracy and higher specificity than PET/CT. Additionally, Reiner et al. [214] obtained a higher accuracy of PET/MRI compared to contrast-enhanced PET/CT, but only in subcentimeter liver lesions without an abnormal FDG uptake.

We found no studies on PET/MRI applicability in ophthalmic oncology. However, a great soft tissue contrast and functional MRI sequences combined with a functional element of PET can provide accurate tumor information at the time of diagnosis and in therapy response evaluation in regions with complex anatomy, such as the ocular space [184]. A low radiation dose is another advantage, and it suggests using this modality while searching for liver metastases or in follow-up imaging. However, some challenges need to be addressed, including the high costs or long acquisition time. Moreover, the research proving PET/MRI superiority over other imaging modalities was studied on only small patient groups, suggesting the need for more research [210].

### 3.3. Contrast Agents for MRI

CA specifically targeting uveal melanoma has not been developed thus far, but attempts are underway to use them differently. One of the studies aims to diagnose Crohn’s disease with a new oral contrast agent (MRI) and compare it to CT imaging (NCT00587210). The efficiency of MR lymphangiography after the administration of ferumoxtran-10 (USPIO) was analyzed in the detection of the neoplasm in pelvic lymph nodes (NCT00147238) [215]. The application of gadoxetate disodium in MRI was compared to CT imaging of hepatocellular carcinoma and cirrhosis (NCT01341132). In other research, the efficiency of MRI with gadofosveset and MR elastography was investigated in prostate cancer diagnostics (NCT01761812).

### 3.4. Hyperpolarized MRI

Although HP MRI is not yet applicable in the diagnosis of uveal melanoma, the studies recently carried out have revealed more and more new applications of that method that may be introduced into clinical practice in the future. The ongoing first-phase study (NCT04772456) focuses on the safety of HP MRI and efficiency of brain tumors (gliomas) imaging after the administration of HP [1-^13^C] pyruvate in comparison to conventional MRI [215]. The visualization of lactate accumulation after the injection of HP ^13^C-Pyruvate in locally advanced cervical cancer (LACC) gives a more specific metabolic characterization of the neoplasm. It is researched to show whether coimaging with DWI and PET could identify areas less sensitive to radiotherapy or brachytherapy using HP MRI (NCT03129776). A lactate level evaluation in tumor cells is also performed in intracranial metastasis treated with stereotactic radiosurgery (NCT03324360). There are also attempts to visualize advanced prostate cancers after the administration of a hyperpolarized marker (^13^C-pyruvate) (NCT04346225) and to control the response to chemotherapy in pancreatic cancer after an injection of HP pyruvate (EudraCT number: 2016-004491-22).

## 4. Conclusions

Uveal melanoma is a rare intraocular malignancy with an unfavorable diagnosis closely connected to the tumor size, time of diagnosis, and presence for distant metastases. The diagnosis of uveal melanoma is usually based on a clinical examination by an ophthalmologist. Ocular ultrasonography is the most commonly used imaging technique that helps in diagnosis, and there is often no need for pathological confirmation before radiation therapy [57,106]. A clinical examination with an indirect ophthalmoscopy and/or fundus photography usually shows the choroidal lesion that is melanotic (55%), amelanotic (15%), or mixed (30%). This lesion is usually associated with orange pigment lipofuscins, which possess autofluorescent properties detected with fundus autofluorescence (FAF) and intrinsic vascularity detected with fluorescein angiography (FFA). This choroidal lesion could be associated with exudative retinal detachment or subretinal fluid as detected by optical coherence tomography (OCT) [106,216,217]. Ocular ultrasonography (B-scan and A-scan) is a noninvasive diagnostic tool that uses ultrasound waves, and it is the most commonly used diagnostic modality for uveal melanoma. B scan can show choroidal tumors that are dome-shaped (75%) or mushroom shaped (20%), with a moderately low internal reflectivity as detected in an A-scan. A B-scan and A-scan can also measure tumor dimensions (base dimension and thickness). Ciliary body melanoma, which affects the anterior part of the eye, can also be evaluated with ultrasound biomicroscopy (UBM) to look at the tumor profile and dimensions [57,106,218]. Posterior uveal melanoma can be confused with many retinal lesions, retinal pigment epithelium, and choroid. The most commonly confused lesions are choroidal nevus, peripheral exudative hemorrhagic chorioretinopathy, congenital retinal pigment epithelium hypertrophy, hemorrhagic retinal detachment or retinal pigment epithelium, circumscribed choroidal hemangioma, and age-related macular degeneration (AMD), respectively [50]. A fine-needle aspiration biopsy (FNAB) of a uveal tumor can be performed when the clinical data are not enough to obtain a definite diagnosis by the ocular oncologist. This biopsy is increasingly performed for the cytogenetic analysis of tumors prognostication [49,57].

SPECT could be used in the initial diagnosis, although exposure to radiopharmaceuticals and the high cost of this method make it more of an additional rather than primary imaging technique [37]. MRI and PET/CT have limited value in the diagnosis of primary tumors; however, they are necessary for tumor staging and the detection of distant metastases [37,38].

The introduction of new advancements in imaging techniques could broaden the horizons in diagnosing uveal melanoma. The development of new radiolabeled radiotracers in PET allows for an extended differential diagnosis and the detection of submillimeter metastases in the liver using melanin-specific radiotracers. Multimodal imaging using PET combined with MRI offers an exceptional soft tissue contrast that can deliver primary tumor characteristics and be used for staging and metastases identification. The visualization of specific metabolic processes is possible with HP MRI, allowing the early detection of uveal melanoma and any metastases. Similarly, new contrast agents for MRI maximize tissue contrast and help to detect intraocular and hepatic lesions. On the other hand, new improvements in noninvasive imaging techniques can also be utilized in personalized medicine, such as theranostics. However, the applicability of those diagnostic advancements has yet to be proven in patients with uveal melanoma; their potential should be tested in future studies.

## Figures and Tables

**Figure 1 cancers-14-03147-f001:**
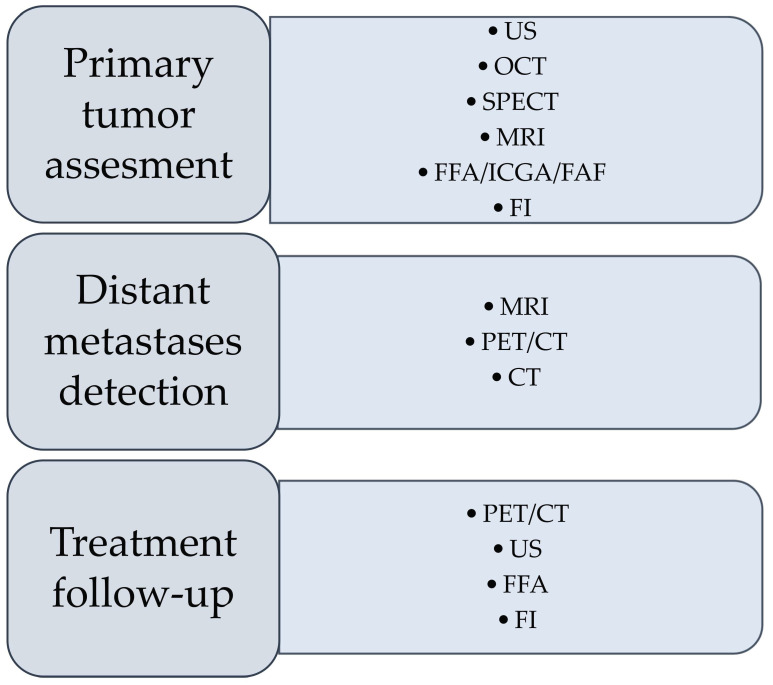
Preferable imaging techniques in different stages of diagnosis and follow-up of uveal melanoma. US—ultrasonography; OCT—optical coherence tomography; SPECT—single-photon emission computed tomography; MRI—magnetic resonance imaging; FFA—fundus fluorescein angiography; ICGA—indocyanine green angiography; FAF—fundus autofluorescence; PET—positron emission tomography; CT—computed tomography.

**Figure 2 cancers-14-03147-f002:**
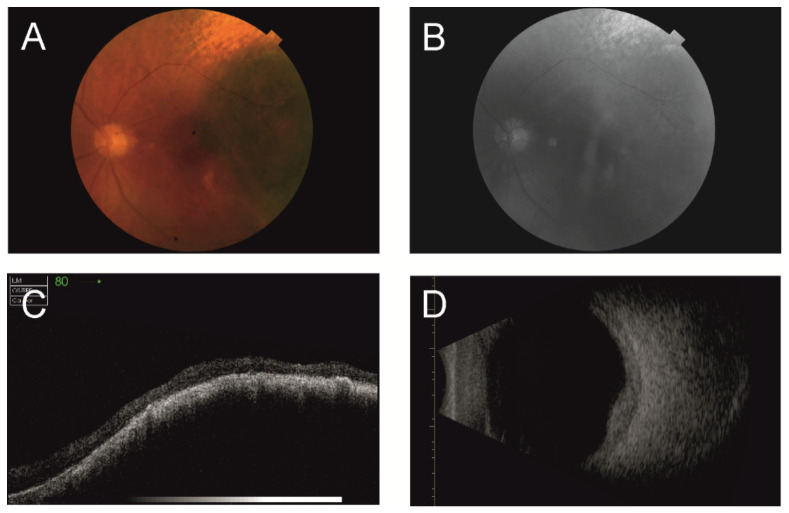
Choroidal melanoma: fundus photography (**A**), red-free image (**B**), OCT scan (**C**), and US (**D**).

**Figure 3 cancers-14-03147-f003:**
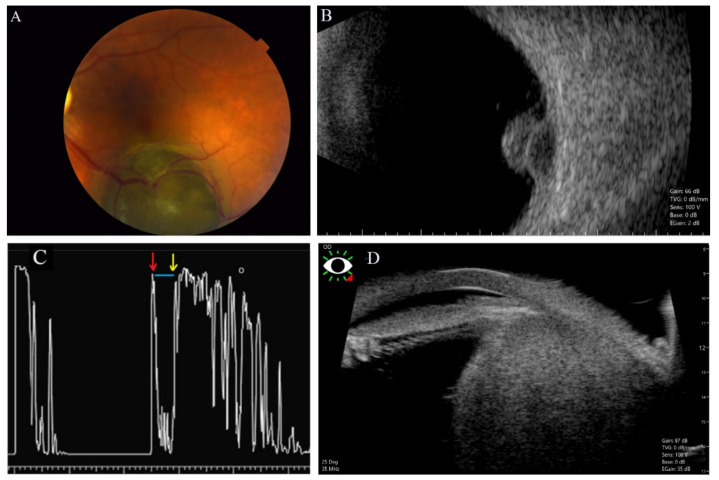
Echographic features of choroidal melanoma. Melanotic choroidal melanoma is mushroom-shaped as detected at the level of inferior arcade of the left eye (**A**). That was shown to be mushroom-shaped with acoustic hollowness in B-scan (**B**). One-dimensional A-scan imaging (8 MHz) for an eye with choroidal melanoma (**C**). The vertical deflections represent echoes from different surfaces in the eye. The red arrow indicates the surface of the retina, the yellow arrow indicates the surface of the sclera, and the interval between them (blue line) shows the low reflectivity of the tumor (choroidal melanoma). Ciliary body melanoma (**D**) detected in ultrasound biomicroscopy (UBM).

**Figure 4 cancers-14-03147-f004:**
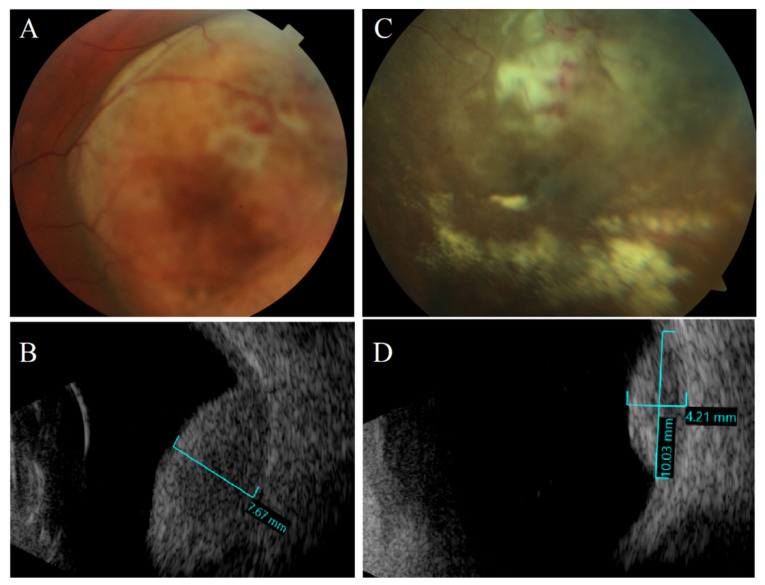
Left eye with amelanotic choroidal melanoma (**A**,**B**) showed a 7.7 mm thick dome shape tumor with moderate low internal reflectivity. Six months after treatment with I-125 radioactive plaque, the tumor showed regression clinically (**C**), and thickness decreased to 4.2 mm (**D**).

**Figure 5 cancers-14-03147-f005:**
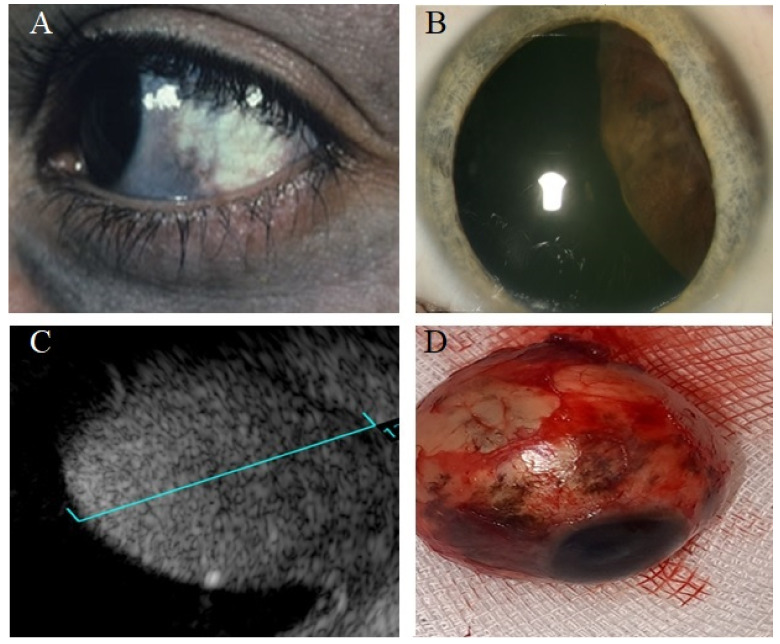
Patient with oculo-dermal melanosis (**A**) was found to have retrolenticular melanotic mass (**B**), which was found to be large choroidal melanoma (13 mm thick) with ciliary body involvement (**C**). This eye was enucleated and shown to have scleral melanosis (**D**).

**Figure 6 cancers-14-03147-f006:**
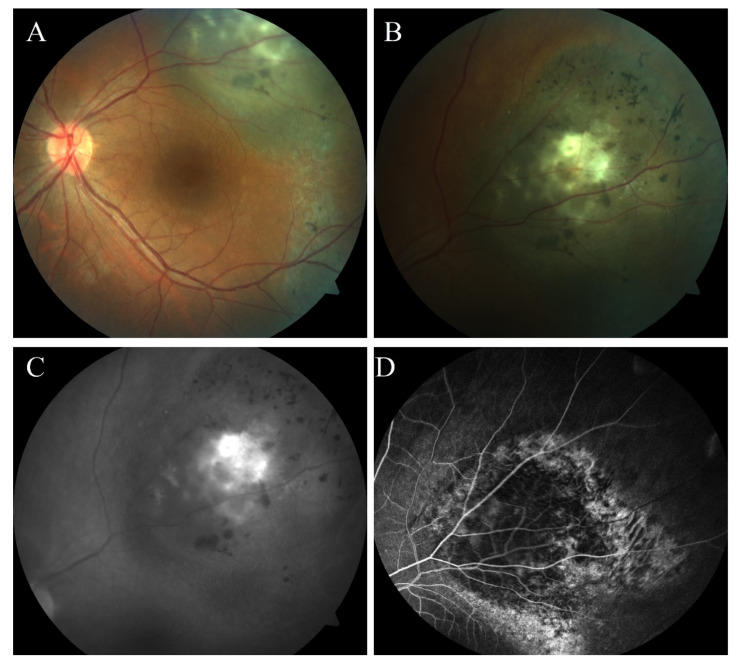
A 30-year-old female with left melanotic choroidal melanoma (**A**). The clinical exam showed RPE hyperplasia and metaplasia, indicating underlying previous choroidal nevus (**B**). The lesion has orange pigment lipofuscins that showed fundus autofluorescence (**C**), and the mass was associated with intrinsic vascularity in FFA (**D**).

**Figure 7 cancers-14-03147-f007:**
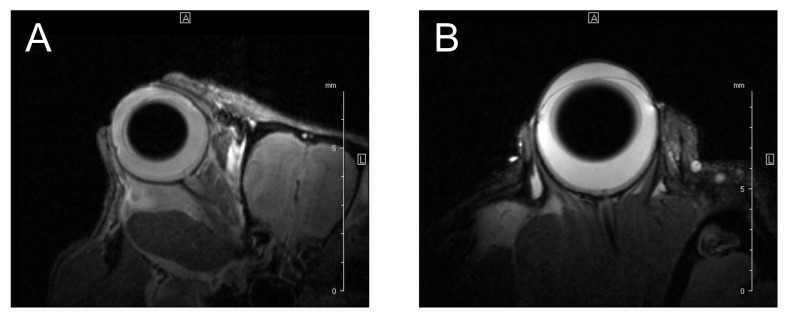
T2-weighted MR images of the eyes of experimental animals—(**A**) mouse and (**B**) rat. High-resolution images enable detailed evaluation of the ocular anatomy.

**Figure 8 cancers-14-03147-f008:**
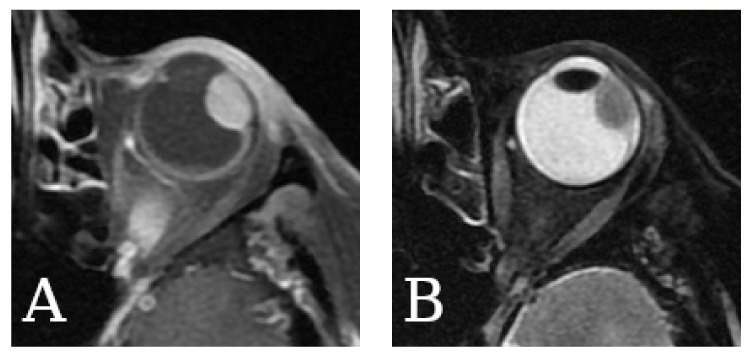
T1- (**A**) and T2-weighted (**B**) MR images of the human eyes with uveal melanoma.

**Table 1 cancers-14-03147-t001:** The summary of advantages and limitations of currently available uveal melanoma imaging techniques.

Imaging Method	Advantages	Limitations	Resolution
US	noninvasive wide availability elementary tool in the initial diagnosis visualization of lesion shape, structure, and vascularization good imaging of retinal detachment, vascular malformations, and hemorrhage good assessment of extrascleral extension and adjacent structures involvement ultrasound biomicroscopy as a developed method in diagnostics relatively low price, between USD 155 and USD 721	operator-dependent method diagnostic difficulties in case of muscle atypical localization or vortex vein enlargement conditions mimicking uveal melanoma such as choroidal naevus, choroidal nevi, metastatic neoplasms, choroidal hemangiomas, disciform lesions, and a choroidal hemorrhages tumors less than 1 mm in thickness could be unnoticed overestimation of the tumor dimension compared to MR	150 and 450 µm, 30 and 60 µm (UBM)
OCT	noninvasive radiation-free high spatial resolution can provide images of tissue microstructure can visualize small lesions in the anterior eye segment low price of around USD 200 on average	small tissue depth that can be visualized not suitable for imaging of pigmented lesions as image shadowing occurs limited use for detection and measurements of posterior eye segment tumors	10 μm or up to 1 μm in UHR-OCT
FFA/ICGA/FAF	noninvasive radiation-free easily accessible possible differential diagnosis use in treatment follow-up (FAF) or assessing the proper placement of radioactive plaque (FAF)	motion artifacts subjective and nonquantitative interpretation of images low diagnostic accuracy when used alone	7.4 to 5 μm
SPECT	noninvasive more sensitive than PET in primary tumor diagnosis suitable for small uveal melanoma diagnosis possible diagnosis in atypical manifestation or ocular complications use of melanic-specific radionuclides (^123^I-IMP) less expensive (USD 1900 on average) and more available than PET/CT	accurate diagnosis is highly time-dependent (up to 48 h) not suitable for amelanotic melanoma diagnosis exposure to radiopharmaceuticals and X-ray radiation with the use of SPECT and CT hybrid	9.3 mm full-width at half maximum (FWHM)
FI	noninvasive, safe, easy to perform, and cost-effective, digital photos can be easily stored and transmitted for consultation	Inability to access deeper layers of the retina, optical aberrations, or cataract influence on image quality, 2D representation of the spatial structure (possible artifacts and peripheral aberrations)	Depends on CCD or phone camera resolution (14 µm per pixel)
CT	high sensitivity, whole-body imaging, a large area covered during acquisition, moderate/high-resolution, less movement artifacts, scanning of a broader range of patients (with metal depositions, pacemaker, etc.), short time of scan (anxiety patients or with claustrophobia)	false positives, radiation exposure, high cost of replacement X-ray tubes, and time-consuming analysis of data	500–625 µm
PET/CT	noninvasive diagnosis of medium and large primary tumors sensitive in the diagnosis of distant metastases, especially in the liver suitable for metastatic disease prognostication can evaluate early treatment response with higher sensitivity than MRI	not widely available risk of falsely positive results in inflammation, infection, and trauma not suitable for diagnosis of small primary tumors exposure to radiation resulting in increased cancer risk very expensive (USD 2227 on average)	PET cameras can provide images with a spatial resolution of approximately 2.4 mm full-width at half maximum (FWHM)
MRI	noninvasive without exposition to ionizing radiation favorable soft tissue contrast a convenient method to visualize intra- and extraocular involvement good to measure the tumor used to detect distant metastases, especially in the liver used in choosing the therapeutic method	difficult to distinguish amelanotic melanoma from melanotic ones choroidal metastases could mimic a uveal melanoma difficult to distinguish tumor from vitreous hemorrhage movement artifacts relatively long duration time of approximately 20 min the relatively high price of EUR 200–EUR 1000	3T MRI 800 µm, 7T 500 to 650 µm, and 32 μm obtained by 9.4T MRI

**Table 2 cancers-14-03147-t002:** Radiopharmaceuticals for PET imaging with potential for use in uveal melanoma diagnosis.

Feature	^18^F-NaF	^18^F-5-FPN	^18^F-MEL050	^18^F-PEG3-FPN
Molecular target	Bone remodeling	Melanin	Melanin	Melanin
Current use	Detection of bone metastases in prostate, breast, lung, and thyroid cancer, and diagnosis of fibrous dysplasia	Preclinical studies detecting small metastases from malignant melanoma in lymph nodes and lungs	Preclinical studies detecting small metastases from malignant melanoma in lymph nodes and lungs	Preclinical studies detecting small metastases from malignant melanoma in lungs and liver
Potential use in uveal melanoma	Detection of bone metastases, assessing extrascleral extension of the primary tumor	Detecting submillimeter metastases in lungs and lymph nodes	Detecting submillimeter metastases in lungs and lymph nodes	Detecting submillimeter metastases in liver and lungs

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
