# Peer review of "Imaging of Uveal Melanoma—Current Standard and Methods in Development"

_cancers, 2022, doi:10.3390/cancers14133147_

Round 1
Reviewer 1 Report
There are significant parts of the paper which are irrelevant to the stated subject (Imaging of uveal melanoma). For example there are several paragraphs devoted to MRI contast agents which do not apply to the subject (pages 21-23). The same applies to the discussion of hyperpolarized MRI (pages 23-25). The Standardized A-scan is not mentioned in the article either in text or by images. This modality is the standard for the ultrasound diagnosis of choroidal melanoma as used in the Collaborative Ocular Melanoma Study (Arch Opthal. 2003 Aug;121(8):1163-71). This is the major imaging modality used in major ocular oncology centers and the others addressed in the article (FFA, OCT, FA etc) are rarely necessary in the diagnosis of choroidal melanoma. There are a number of grammatical and spelling errors which require editing by someone conversant with English.
Author Response
Thank you for your suggestion. We have put the responses in the attached file.

Reviewer 2 Report
I consider that the topic is a very good one and the writing of the article is very well structured.
Also, I would like to be a correlation with the same pathology in Black or Asian patients.
The diagnosis of uveal melanoma is fundamentally based on clinical evaluation, however the research asks an interesting question, in order to confirm the diagnosis, what is the potential of new advancements in imaging techniques?
I consider the article relevant in the field because a detailed knowledge of imaging used in uveal melanoma may provide an important contribution in the diagnostic workup. The paper provides an update on the main current and future imaging techniques but also a comparison of these methods.
There are no specific improvements to be done.
The conclusions are consistent with the purpose of this review in order to provide the radiologists with awareness about diagnostic methods. In the first section authors sumarize US, OCT, FI, FFA, MR, CT, SPECT and PET of the eye with clear images of the pathology details that are completed with newest experimental techniques with the potential in diagnostics.
The references are among the newest and most relevant in the field and the paper comes with a rich paraclinical imagery that completes the information presented in the text.
Author Response
Thank you for your comments. We have put the responses in the attached file.

Reviewer 3 Report
In the manuscript "Imaging of uveal melanoma – current standard and methods in development" by MaÅ‚gorzata Solnik et al., the authors attempt to perform a comprehensive review about the imaging methods to diagnose and perform the follow-up of patients with uveal melanoma. In addition, the authors highlight some promising new imaging methods that are currently being developed. The theme is interesting and extremely relevant for the field. Even though it is a bit long, the paper is well-written and organized. However, there are some aspects that should be improved and merit the attention of the authors, namely:
- The introduction addresses different clinical, epidemiological and prognostic factors on uveal melanoma in a somehow confusing manner. Some important aspects are missing and others are not accurately described. In addition, some key recent papers are not cited:
- Martine J Jager et al.; Uveal Melanoma. Nature Reviews Disease Primers, 2020.
- Lamas et al.; Prognostic Biomarkers in Uveal Melanoma: The Status Quo, Recent Advances and Future Directions. Cancers, 2021.
- The sentences “Genetic factors such as chromosome 3, 1p loss, or 8p gain correlate with increased risk for metastasis [1,23]. Tumors are dominated by epithelioid cells with high mitotic activity” on page 3 have elements of innacuracy and should be improved.
- Researchers at Wills Eye Hospital have extensively studied risk factors for choroidal nevus transformation into melanoma. In a recent paper (Carol L Shields et al. Small choroidal melanoma: detection with multimodal imaging and management with plaque radiotherapy or AU-011 nanoparticle therapy. Current Opinion in Ophthalmology, 2019) they proposed that the mnemonic ‘To Find Small Ocular Melanoma Doing IMaging’ (TFSOM-DIM) is extremely useful assisting medical doctors in the detection of factors that increase the risk of choroidal nevi to transform into small choroidal melanomas. To further highlight the role of imaging in the diagnosis of uveal melanoma, the paper should describe this mnemonic and describe key studies linked to it.
- The paper would benefit from a small discussion on the best combination of methods to more accurately diagnose uveal melanoma using imaging methods.
- Throughout the paper there are small typos that need to be corrected. For example, in the legend of Figure 2.
Therefore, the present review paper summarizes relevant infomation about imaging in uveal melanona, but it will be mandatory to perform additional changes in the present manuscript if it is considered relevant for publication in Cancers.
Author Response
Thank you for your suggestions. We have put the responses in the attached file.

Round 2
Reviewer 1 Report
1. There are some minor grammatical constructions that need revision.
Introduction first paragraph: "In the anterior, tumor usually localize in the uveal tract that contains the iris. In the posterior, the tumor may localize the choroid and ciliary body."
Page 3 first paragraph: " ...very low in African and American."
Page 5 second paragraph: "...using the fixation mentioned methods allows determining the probability transformation of choroidal nevi into melanoma."
2. Reference improperly numbered.
Page 6 last paragraph: the reference to the collaborate ocular melanoma study [5,6] doesn't correspond to the reference number in the section listing references
3. A-scan
Page 6 last paragraph: the authors mention the importance of A-scan as developed by Ossoinig [standardized A-scan] but the image shown in figure 3 is not a standardized A-scan but an A-scan derived from the B scan envelope. This is confusing and should be replaced by a standardized A-scan image of a choroidal melanoma
4. Confusion of hyperreflectivity
Page 6 third paragraph: the authors state, "...infiltration is visible as regions of hyperrrefectivity beyond the physiological limits of the sclera..." One of the references they give is the Canadian Journal of Ophthalmology [54] which states, .."characterized extrascleral extension of choroidal melanoma on B-scan ultrasonography as an area of relative echolucency immediately behind the sclera.." This would correspond to relatively low reflectivity on the A-scan.
5. Differential diagnosis of choroidal melanoma by ultrasound
Page 8 second paragraph. The authors give the differential diagnosis of melanoma by ultrasound to include choroidal nevus, retinal hamartoma, tuberculoma, or neurilemma. Most of these are quite rare. A more accurate list is that mentioned in the book by Byrne and Green: choroidal nevi, metastatic neoplasms, choroidal hemangiomas, disciform lesions and a choroidal hemorrhages (Ultrasound of the Eye and Orbit. St Louis. Mosby;2002)
Author Response
Dear Reviewer,
Thank you very much for the additional comments and suggestions. Please see the attachment.

Reviewer 3 Report
In the revised manuscript "Imaging of uveal melanoma – current standard and methods in development" by MaÅ‚gorzata Solnik et al. the authors addressed in a satisfactory manner all my previous concerns and, thus, I have no further objections towards the publication of the manuscript in Cancers.
Author Response
Dear Reviewer,
We would like to thank you for taking the necessary time and effort to review the manuscript. We sincerely appreciate all your valuable comments and suggestions, which helped us in improving the quality of the manuscript.